



# Hydrological corridors for landscape and climate restoration: Prioritization of re-greening areas in Kenya and Tanzania

Judith E.M. Klostermann[1], Luuk Fleskens[1], Erik Querner[1], Herbert Ter Maat[1], Ronald Hutjes[1], Fons Jaspers[2], Sander de Haas[2]

[1] Alterra, Wageningen University and Research, Wageningen, 6700AA, The Netherlands
[2] Naga Foundation, Amsterdam, 1012KL, The Netherlands

*Correspondence to*: Judith E.M. Klostermann (judith.klostermann@wur.nl)

**Abstract.** The Naga Foundation aims to implement durable re-greening interventions to increase local soil sustainability and regional water availability. When this is done on a large enough scale such landscape changes may also lead to positive regional climate impacts. Naga is developing a plan to re-green 15 large areas in Eastern Africa, creating a so-called hydrological corridor. Four potential hydrological corridors have been identified in Kenya and Tanzania, all four of them around Mount Kilimanjaro. To select the most promising corridor, a method was developed to support a decision in a situation where few data are available. The method is based on maps, models and literature from four different disciplines concerning soil, water, climate and social institutions. The findings favour the Tanzanian corridors and especially the Tanzania-East one, to start with re-greening projects. In that region many applicable land management options combine with a high potential for restoring soil organic matter, the highest rainfall recycling potential exists in the more favourable long rains season, while finally also the Tanzanian governments both at national and at local level seem more dependable for supporting hydrological corridor implementation.

Keywords: Re-greening, hydrological corridors, land degradation, sustainable land management, climate feedbacks, adaptive capacity, Tanzania, Kenya

## 1. Introduction

Worldwide, drylands are increasingly affected by desertification. A complex interplay of resource exploitation, population increase and environmental change is driving land degradation processes with both local and off-site impacts that may culminate in desertification, defined as the loss of productive capacity of drylands (UNCCD, 1994). Land use and climate change, over-abstraction of (ground-) water resources, and unsustainable land management are the direct driving forces of land degradation, but are in turn caused or exacerbated by indirect driving forces such as food insecurity, poverty, short-term management perspectives, and breakdown of governance institutions (Geist and Lambin, 2004; d'Odorico et al., 2013; Bisaro et al., 2014). Addressing such root causes is difficult and can only succeed when stakeholders at all levels are aware



of the extent of impacts and support options to restore degraded lands (Reynolds et al., 2007; Bisaro et al., 2014; Fleskens and Stringer, 2014).

Desertification includes the reduction of evaporating green cropped area, causing also higher temperatures and dryer air (Stroosnijder, 2009). In the atmosphere this may influence the weather features causing longer dry periods before reaching the higher critical saturation level for precipitation again.

Vegetation degradation may affect local and regional climate by reducing the capability to recycle rainfall, but the reverse may also be true (Pielke et al. 2007, Ter Maat et al. 2006). Re-greening on a sufficiently large scale, with shrubs and trees with roots deep enough to sustain green leafs longer into the dry season than grasses can, will increase evapotranspiration. This enhanced atmospheric moistening may lead to additional cloud formation and rainfall generation, especially in the aftermath of the rainy season and when helped by orographic lifting, for example, near the Kilimanjaro.

The Netherlands-based Naga foundation aims to implement re-greening interventions over large areas to increase local soil sustainability, regional water availability and larger scale atmospheric effects. In this vision, rainwater harvesting is used to restore degraded lands and to increase soil moisture and water availability. This should lead to an increase of vegetation in degraded lands; the vegetation will retain fertile soils, slow down runoff and improve infiltration (Rockström et al., 2010; Rockström and Falkenmark, 2015). When this is done on a large enough scale the landscape changes also may affect local climatic conditions (evapotranspiration, temperature, cloud formation), which in turn can lead to positive regional climate impacts (Borst et al., 2015). It is supposed that water masses (oceans and seas), land elevations and predominant wind directions like a monsoon also play an important role in determining the rainfall regime (Pielke 2001; Pielke 2007; Taylor 2007). Apart from restoring organic carbon, re-greening is expected to have spin-offs like reduced flooding and river erosion, improvement of agricultural land, higher groundwater levels and a more balanced base-flow (Borst et al., 2015).

To achieve re-greening of degraded lands Naga intends to re-green 10 to 15 large areas in Eastern Africa, each 20 km2, within a so-called Hydrological Corridor of 30,000 to 85,000 km2. Four potential Hydrological Corridors were roughly drawn in Kenya and Tanzania, all four of them around Mount Kilimanjaro because its slopes promise upward winds which are needed for a local climatic effect (see Figure 1). Two of the potential corridors range from Mount Kilimanjaro eastwards to the Indian Ocean: one is located in Kenya (Ke-E), the other one is mainly in Tanzania (Tz-E). The other two corridors range from Mount Kilimanjaro westwards to Lake Victoria: again one in Kenya (Ke-W), the other mainly in Tanzania (Tz-W).



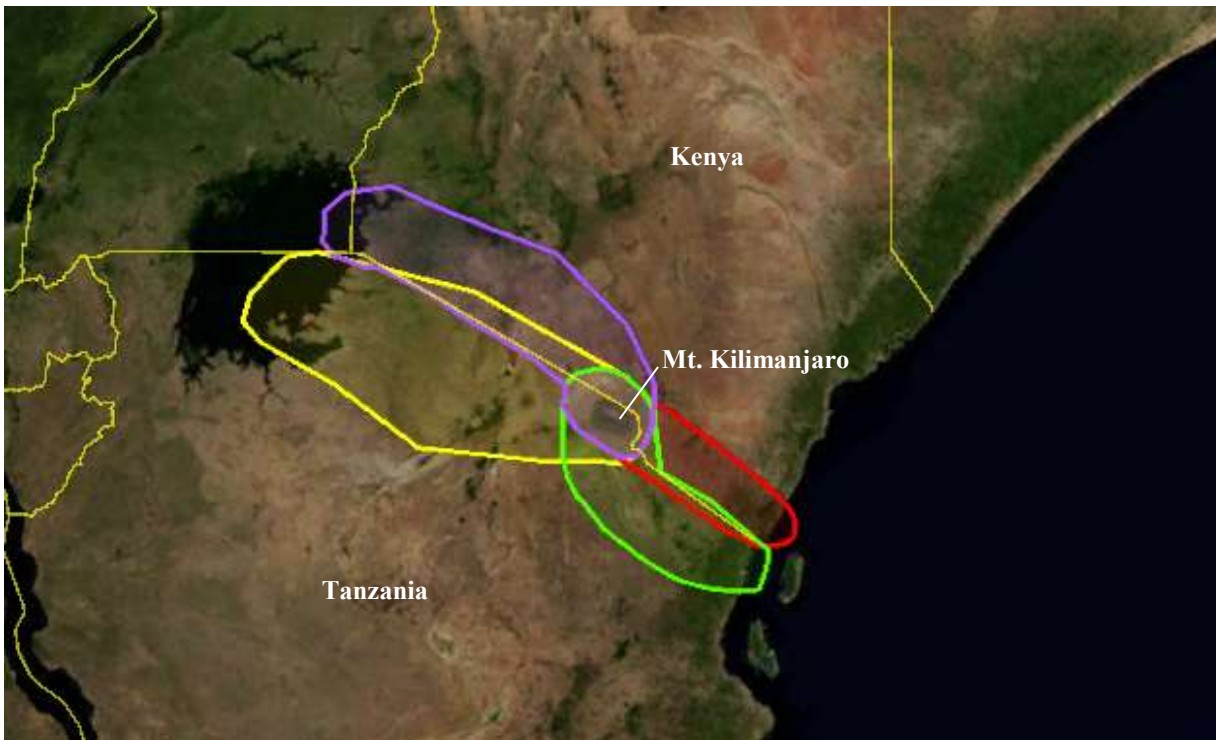

**Figure 1: Potential Hydrological Corridors in Kenya and Tanzania (imagery from Google Earth)**

The aim of this article is to develop a method supporting a decision in a situation where few data are available, based on
maps, models and literature from four different disciplines. First we determine which of the four potential corridors has the
highest potential for sustainable land restoration, based on soil and hydrological information. Next we investigate which
areas within such a corridor hold the best potential for altering the local and regional climate. Furthermore the potential of
the social and institutional context for supporting the required land use changes is assessed.

## 2. Method

We used a number of objective criteria for prioritizing potential areas for re-greening interventions. Important characteristics
for the re-greening potential are land degradation status, soil characteristics, drainage characteristics, physical potential for
restoration, possible climate effects and institutional potential for restoration (Borst et al., 2015). Each of these aspects can
be further subdivided into variables that are worth considering. In Table 1 we have selected a concise set of variables that
can be measured or modelled for a first, rough prioritization which areas are the most promising as part of this study. The
table also shows the motivation for selecting each indicator, a brief summary of the methods chosen, and the a priori
assumption when an indicator is in favour of selecting an area.




**Table 1: Selected indicators, motivation and summary of analysis method.**

| Relevant aspect | Indicator | Motivation and method | Assumption when an outcome is positive for selecting an area |
|---|---|---|---|
| Degradation status | Soil depth loss | Soil erosion is one of the most important processes of dryland degradation. Soil depth loss is an indicator of soil erosion and results in reduced productivity (De la Rosa et al., 2000) | Where the ongoing rate of soil depth loss is highest, there is significant scope for soil conservation to reverse land degradation |
| | Soil organic carbon (SOC) restoration potential | SOC restoration directly contributes to climate change mitigation but also leads to direct effects on crop yields (Lal, 2004) | Given a continuation of current land use, soils with highest historical or ongoing SOC loss present the highest potential for SOC restoration |
| | Number of relevant Sustainable Land management (SLM) categories | A diversity of applicable SLM categories enables land managers to choose between multiple options, enhancing chances that one of them is viable from a utility point of view (Fleskens et al., 2014) | More choice is better. In addition, the co-benefits of SLM measures can be better served by implementing a variety of measures |
| Hydrological restoration potential | Slope | Slope influences infiltration-runoff ratios (for groundwater recharge). Slope can also affect the number of suitable SLM options and cost of implementation (Fleskens, 2012) | Plains are better for groundwater infiltration than (steep) slopes |
| | Sand fraction | In combination with the slope and daily rain rate, the sand fraction will give an indication how much erosion is likely to happen in a region (Angima et al., 2003) | A smaller sand fraction gives an indication of less soil erosion. |
| | Daily rain rate and rain sum (5 days) | The daily rain and the rain sum of 5 days gives the potential source of water to harvest (Kimani et al., 2015) | An increase in rainfall and particular the higher sum of rainfall over 5 days is beneficial for re-greening. |
| | Presence of small and large dams | The ability to construct small or large dams depends on numerous factors, like upstream size of the catchment and rainfall amount (FAO, 2010) | Larger catchments mean favourable conditions for water retention and storage in the river. |
| Climate feedback effects | Precipitation/ climate model sensitivity to re-greening | By using various options of land use scenarios the sensitivity of the local weather is calculated in the transient season | An increase in rainfall for the area of interest |
| Institutional potential for | Adaptive capacity of national | The Adaptive capacity wheel provides a quick assessment of many governance aspects (Gupta | Higher adaptive capacity leads to a higher potential for re-greening |



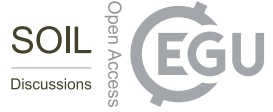

| restoration | institutions | et al., 2010). National institutions are relatively easy to map based on available literature. | projects |
|---|---|---|---|

After the analysis of each indicator a method is needed to add up the 'apples and oranges' in a transparent way, to come to a final conclusion. We developed a set of weighting and aggregation rules for a semi-quantitative overall assessment and ranking of map units in terms of suitability and priority for restoration projects. Giving weight to the various criteria ideally

should be discussed with all relevant stakeholders, but this was beyond the scope of the reported research project. The weighting is now based on the expert judgement of the authors only.

## 3. Existing land degradation in terms of soil loss, soil carbon and options for recovery

### 3.1 Method for soil restoration

For prioritizing regions for soil restoration, we take the existing level of degradation as a relevant first indicator. Degradation

can be assessed by the soil depth loss and the carbon content of the soil relative to undisturbed conditions. Different sustainable land management interventions have different applicability limitations and their effectiveness also depends on environmental conditions. The number of options for recovery and their simulated effectiveness represent relevant indicators to include in our study.

In order to identify areas with the largest potential for restoration and to choose the most optimal measures to effectuate this

we follow a mapping approach. We use the WOCAT database (www.wocat.net) to identify the effects of different categories of restoration options or sustainable land management (SLM) measures.

The mapping approach is built on the premise that SLM measures can affect a degraded soil in two ways:

- Restoring soil organic carbon (SOC). The SOC content of degraded land can be restored by improving soil cover and enhancing soil health, either through amendment of chemical soil fertility (adding manure, composting) or

through altering the physical or biological properties of soils (mostly indirectly).

- Preventing soil loss. Soil loss through erosion processes can be controlled by reducing the susceptibility of soils to the impact of rain and wind, or by limiting the transport capacity of these vectors.

Most SLM measures contribute to both effects simultaneously. In order to calculate the effects, both restoration and

prevention were expressed regarding their effect on SOC, which can be considered a proxy indicator for soil productivity. The extent to which they contribute depends on time.

However, the shape of the restoration and prevention trend lines is governed by a number of factors (Hutjes et al., 2016):

- Time after investment; A literature survey on results from multi-year and long-term field experiments of restoration options showed that the most promising technologies tested achieve a logarithmic increase of SOC over time, i.e.



rapid initial gain that flattens off over time. This logarithmic curve was generalised with the aid of WOCAT expert data.

- Maximum potential for SOC restoration; this provides ceiling values for SOC content based on given current land use or natural conditions, based on finding analogues for each situation. The maximum potential for SOC restoration was taken as the difference between current and maximum SOC in soils.

- Remaining soil depth for soil loss prevention; scenario analyses provided information on the diminution of topsoil depth over time due to soil erosion processes. Topsoil loss is considered linearly in this study.

When implementing the approach above spatially, there are three possible situations:

1. If there is no restoration potential in a given grid cell, no SOC increase will be possible; note that for SLM technologies the potential is defined as keeping the same land use, whereas for reforestation options the natural potential is used as maximum value.

2. As long as restoration is possible and the potential is not reached, for SLM technologies SOC restoration is defined in proportion to the generic SOC restoration curve, with initial SOC stock, an expert estimate of the % SOC increase, and assessment lag determining the level of SOC increase.

3. When the potential level of restoration is reached, no further increase of the SOC is possible.

SLM measures were selected from the WOCAT Technologies database (www.wocat.net). The database in total features over 400 individual SLM measures originating from a variety of countries worldwide. Individual measures were selected for analysis based on the criterion that they included an estimate of the effect on SOC content of soils. This yielded a set of about 200 technologies, which was further delimited based on availability of realistic investment cost information. Based on this dataset, for the purpose of a global assessment of SOC restoration potential a classification of SLM measures was needed that a) was sufficiently refined to meaningfully distinguish clusters of SLM measures that have similar applicability limitations, costs and effect; b) offered spatially differentiated potentials for restoration actions; and c) included sufficient member technologies to characterise costs and effect of the category. The resulting classification of SLM categories is summarized in Appendix A.

**3.2 Results for soil restoration**

Figure 2 shows the annual soil depth loss in the research area, whereas Figure 3 shows the SOC restoration potential. Soil depth loss is predominantly confined to areas with steep slopes, e.g. the foot slopes of Mount Kilimanjaro, the Rift Valley in Ke-W and the Usambara Mountains in Tz-E. SOC restoration potential seems particularly elevated in Tz-E. The area concerned is to a large extent in use as rain-fed farmland which may explain a low current SOC content relative to the potential SOC content.




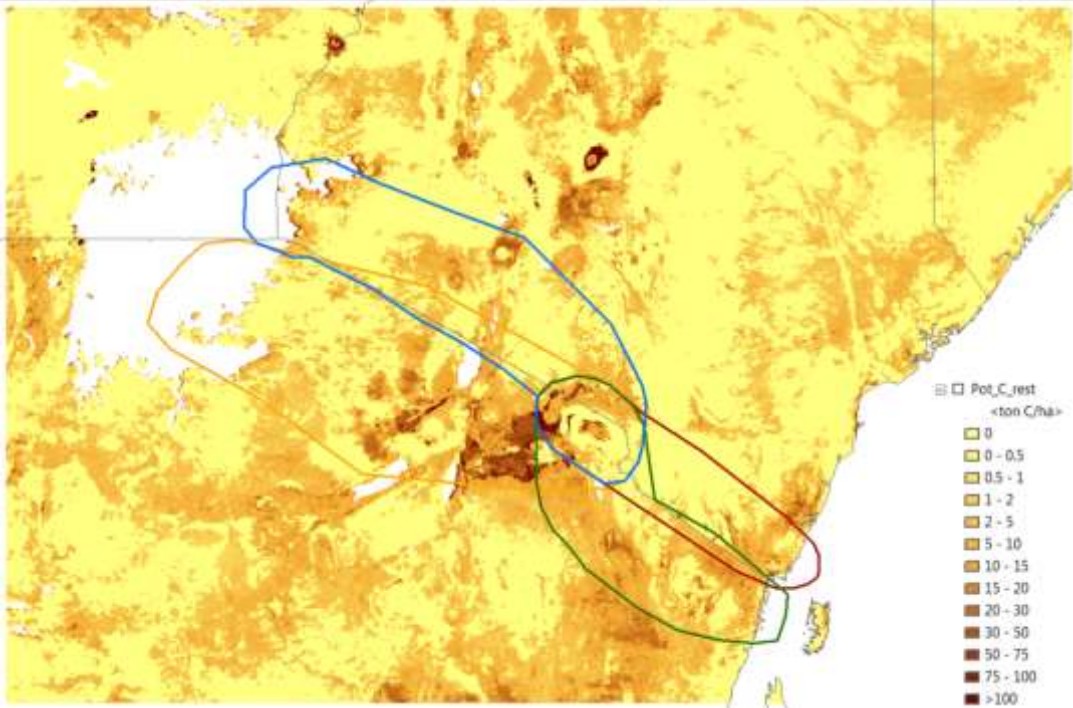

**Figure 2: Annual soil depth loss (mm/year).**

**Figure 3: SOC restoration potential (ton C/ha).**




Figure 4 shows how many categories of measures can be implemented in each location. Most of the corridors show potential for several types of measures on most of the territory. An exception is Ke-E where a significant proportion of the area is either not degraded and in need of restoration (as supported by Figure 3 and 4), or no measures are applicable. Overall, Tz-E is the area where, on average, the highest variety of SLM measures is applicable. Figure 5 shows the potential for effective SOC restoration using the portfolio of SLM measures assessed. The pattern mostly reiterates where the highest SOC gains can be achieved (Figure 3), but also shows some areas where erosion control can make an effective contribution (e.g. Tz-E, cf. Figure 5).

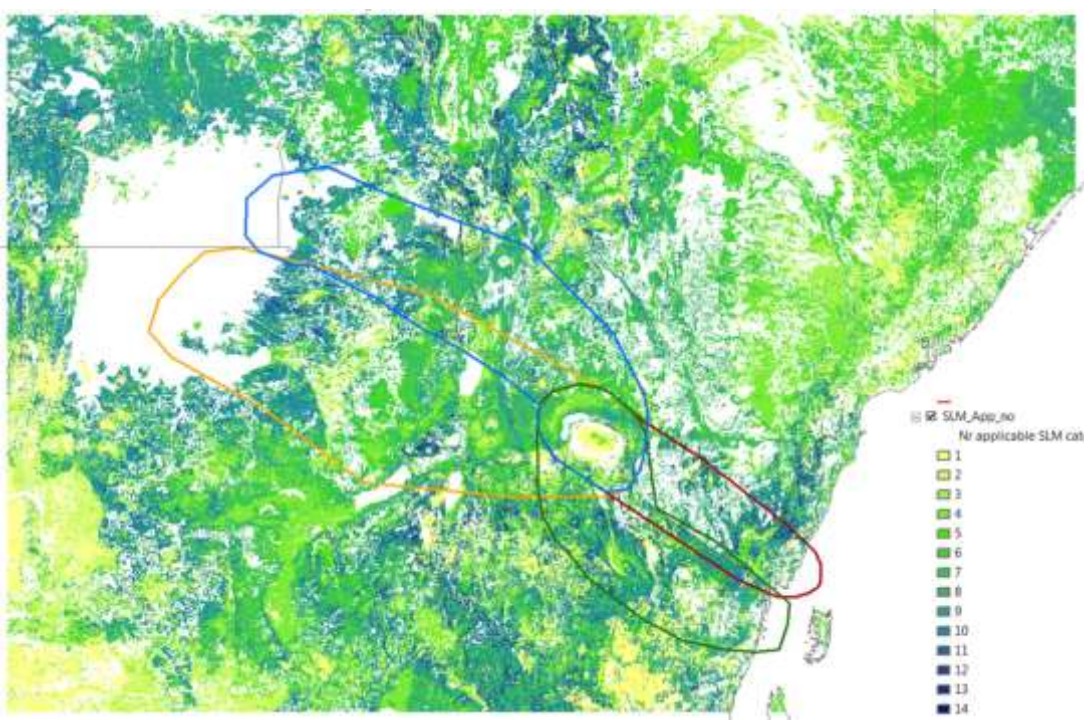

**Figure 4: Number of applicable SLM categories**





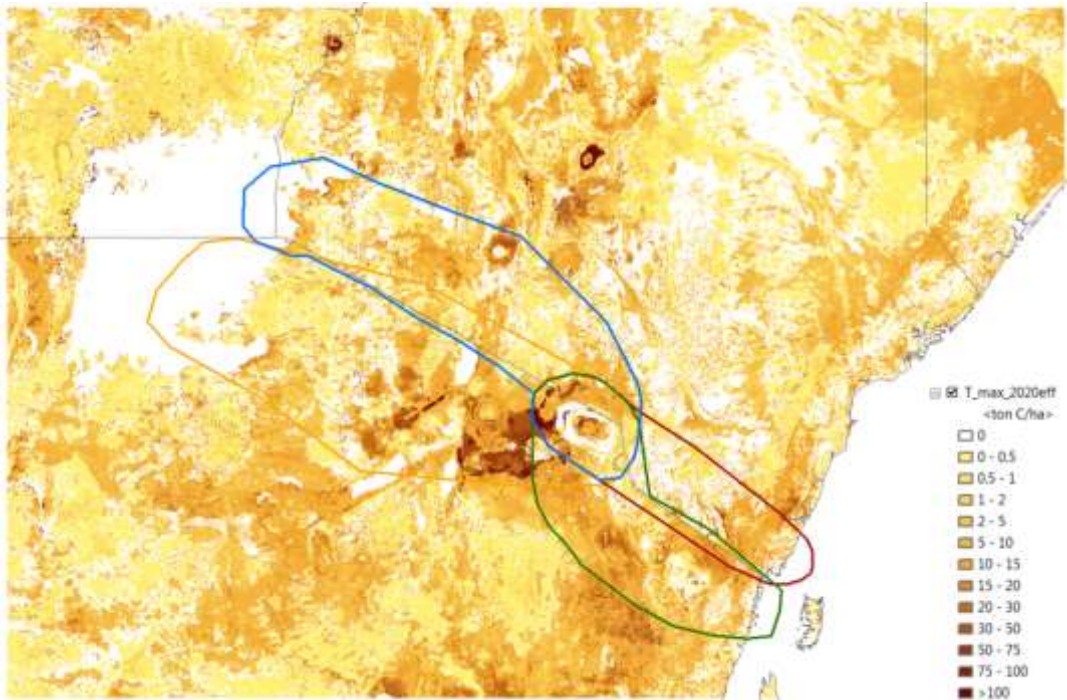

**Figure 5: Total effective SOC restoration potential by 2020.**

Figure 6 schematically shows the cumulative scope and effective SOC restoration potential. Ke-E has a high total restoration
scope, but a relatively low potential for effective restoration using SLM measures. Tz-W and Tz-E also have a quite high
scope for SOC restoration, and they feature the highest potential in terms of achievable restoration potential using the SLM
measures considered. Ke-W scores lower on both accounts.



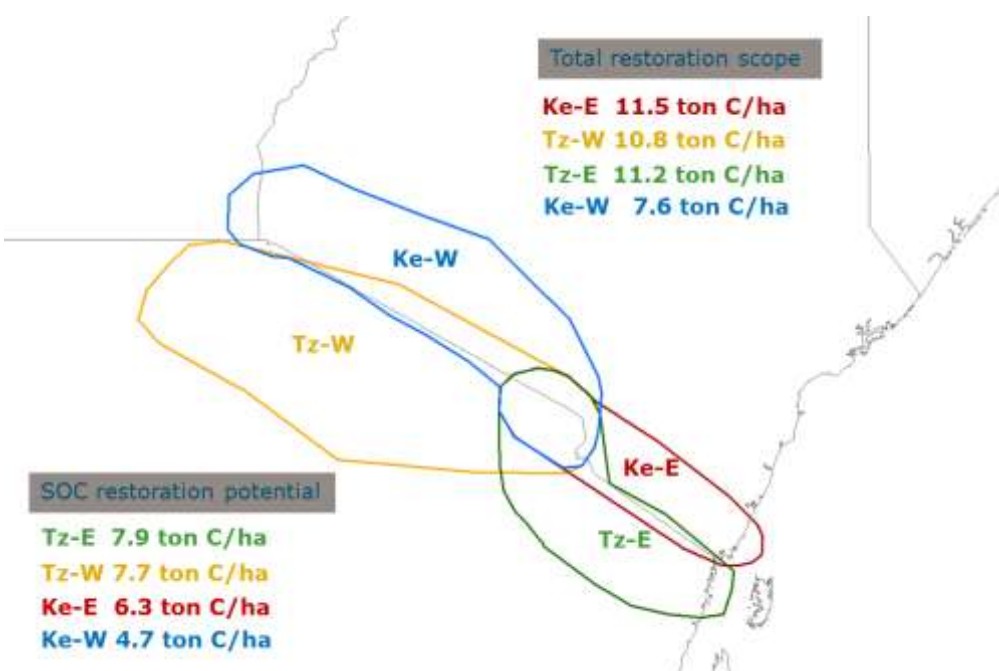

**Figure 6: Cumulative SOC restoration scope and potential (average per potential hydrological corridor).**

## 4. Hydrological analysis of downstream erosion, groundwater flow, and river water flow and quality.

### 4.1 Method for hydrological analysis

5    In sub-Sahara Africa, water availability is often insufficient and irregular. Furthermore, flash floods frequently occur, causing soil erosion. For the purpose of re-greening water runoff must be captured in the soil or enabled to infiltrate towards deeper groundwater layers. Such natural retention measures are needed at specific locations for optimal retention. From a hydrological point of view we search for areas with either the greatest natural water retention capacity or areas where the water retention can be augmented by interventions.

10   In a search for the best locations for retention measures to be implemented in the field, we first considered an integrated GIS assessment using available data for (parts of) Kenya and Tanzania. We used digital elevation maps and derived from them the ground slope, extent of rivers and sub basins. Combined with local rainfall and evapotranspiration data this gave information on water quantities at grid scale. These data were combined with land use and soil data.

The analytical approach was based on a decision to use easily available digital data and a grid cell size of 250x250 m, since
15   that cell size guaranteed enough detail at the local and regional scales. From the soil data spatial information was acquired on soil moisture retention characteristics and the infiltration rate.

In the process of degradation, the ground surface has often become impermeable, which can severely reduce the infiltration capacity of the soil. Important is that more water is stored in the upper soil (or root zone) or seeps to deeper layers. Such a



flow to deeper layers is beneficial for the recharge of groundwater reservoirs that eventually will drain downstream in the basin and increase the base flow. Alternatively, groundwater can be utilised as drinking or irrigation water by pumping up this water. In addition, the reduction of surface runoff towards streams helps to reduce soil erosion, the formation of local flash floods, and damage to river infrastructure.

Different water storage abilities and infiltration measures need to be considered as well as landscape situations that are suitable for the retention and infiltration of water. The main factors influencing the storage are ground slope, infiltration capacity, land use, variation in rainfall intensity, and the surface water infrastructure of river basins and sub-basins. Favourable water retention and infiltration measures were identified and were given weights against these important factors. To that end a spatial database was built and information was collected on relevant properties that could be visualised,

combined and weighted.

For the spatial hydrological analysis, the following data were used:

- Digital elevation model (DEM) and derived data of the surface slope and surface water network (basins and sub-basins, depressions, etc.);

- Soil map and derived data on infiltration capacity; here based on ISRIC AFSOILGRID250

- Rainfall data and derived data on maximum intensities, variation, etc. based on the Watch Forcing Data (Weedon et al., 2011).

From the DEM a drainage map has been constructed that not only shows the (most probable) location of (intermittent)

streams, but that can also be used to calculate the upstream area of the catchment above any point along that stream. This type of map can be used to identify potential locations of small dams for the reduction of stream flow and erosion and the enhancement of storage and infiltration and aquifer recharge.

Next several of these maps can be combined, by scoring each theme. For each theme, the desired characteristics are given a higher score compared to less favourable ones. Table 2 is a first try at such a ranking of desired characteristics.

On steeper slopes the chances for infiltration of rain are generally reduced in favour of runoff. Therefore, steeper slopes were given a lower score, in the example perhaps a bit too conservative. Coarser soils (larger sand fraction) have a higher infiltration capacity, thus if ground water recharge is a project objective sandy areas are given a higher score. For recharge to occur rain is needed, so areas with sufficient rain are given a higher score. However, very high rain intensities could lead to infiltration capacity exceedance and the formation of runoff and erosion.

Thus each theme can be scored and combined with others, depending on the objectives. Obviously there is some arbitrariness in doing this and scores should be subject to discussion. The scoring needs to be modified depending on the objectives of the project and in fact such discussions can help to make choices more explicit and to better clarify the objectives.



**Table 2: Overview of scores per theme considered for hydrological analysis.**

| | DEM | Soil map | Meteorology | | Upstream catchment | |
|---|---|---|---|---|---|---|
| Initial score | slope | sand fraction | daily rain rate Febr-May | rain sum 5 consecutive days | small dams | large dams |
| 0 | >1.0% | <0.35 | <1mm | <30mm | <100 or >1500 km$^2$ | <400 or > 5000 km$^2$ |
| 1 | 0.5-1.0% | 0.35-0.45 | 1-2mm | 30-40mm | 1000-1500 km$^2$ | |
| 2 | | 0.45-0.55 | 2-3mm | | 100-200 km$^2$ | 400-1000 km$^2$ |
| 3 | 0.25-0.5% | 0.55-0.65 | 3-4mm | 40-50mm | 700-1000 km$^2$ | 2500-5000 km$^2$ |
| 4 | | >0.65 | 4-5mm | 50-60mm | 400-700 km$^2$ | 1000-1500 km$^2$ |
| 5 | 0-0.25% | | >5mm | >60mm | 200-400 km$^2$ | |

## 4.2 Results of the hydrological analysis

Figure 7 shows the range in scores for combining slope, soil and average daily rainfall (period Feb – May). The range in

5   scores is between 0 and 14. The darker (more purple) areas are the more sandy areas that receive rainfall in this period. The

figure suggests the highest potential is in Tz-E while the lowest is in Ke-W.

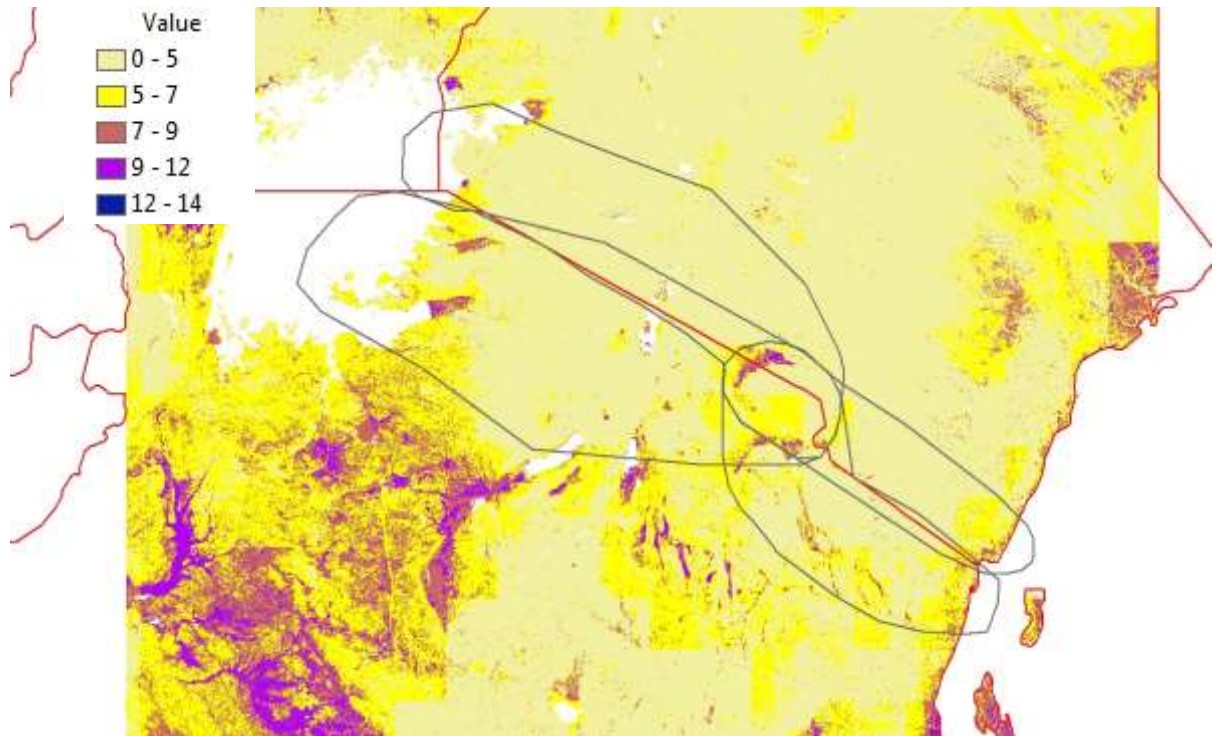

**Figure 7: Range in scores for hydrological potential combining slope, soil and average daily rainfall. The range in score is between**
10   **0 and 14.**





## 5. Feedbacks to the local climate change: temperature, rainfall and wind.

### 5.1 Method regarding climate impact

One of the novelties of the Hydrological Corridor concept is that it not only aims to improve vegetation and hydrological conditions on the ground, but that it also leads to positive effects on local climate. Feedback effects of increased evaporation

from a greener land surface on rainfall can be expected in the transitional periods from dry to wet seasons and vice versa (Findell and Eltahir 2003, Tuinenburg, Hutjes et al. 2010). In the midst of the rainy season the atmosphere is so moist (advected from the oceans) that rain will occur irrespective of local evaporation. In the midst of the dry season the atmosphere is so dry that local evaporation will not raise atmospheric humidity to condensation levels. In the transition periods, when atmospheric humidity is just below a threshold initiating rainfall generation and/or convection, the local

evaporation may increase to just above the threshold and cause positive feedback to further rain and evaporation.

Normally, prior to the rainy season there is no soil moisture to sustain any evaporation, except in irrigated areas. Since Naga aims to increase soil moisture and vegetation through rain harvesting and retention techniques with an effect after the rainy season but not before, we will not consider this period. At the end of the rainy season and into the early dry season there usually is enough soil moisture left to sustain evaporation. We may expect to see the first climate improvements from re-

greening, if any, occurring in this period. Our analysis thus targets these periods.

Positive effects on rainfall can only occur when the enhanced atmospheric humidity from local evaporation is lifted to condensation levels where cloud formation and rain generation can occur. However, the same evaporation that adds moisture to the atmosphere will also cool it, thus reducing buoyancy of the air, reducing boundary layer growth and thus preventing the -albeit humidified- air from reaching condensation levels. Thus, in flat areas the effect on precipitation is the result of

two counter-active processes (humidifying, cooling), leading to rather uncertain overall effects. If the humidified air is blown against rises in the land (hills, mountains) the resulting orographic lifting may bring the humidified air to condensation levels despite its reduced buoyancy. Thus we expect higher chances of climate improvements from re-greening when the land rises downwind of the greener areas. Indicators to identify the proper region for a Hydrologic Corridor may additionally be orientation with respect to predominant monsoonal winds.

We follow two tracks to analyse these processes for the Kenya-Tanzania border area, where the topography around Mt Kilimanjaro may increase the chance of positive climate feedbacks from greening:

- First the climatology of the area, the dominant wind patterns throughout the year and associated temperature, humidity and rain distribution are analysed;
- Next sensitivity experiments are executed with a regional weather model, where greener vegetation is forced within

30        the four corridors initially considered.





## 5.2 Analysis of the Mt Kilimanjaro regional climatology

Using several global data sets (Hutjes et al., 2016) available at daily, 0.5° resolution an analysis has been made of monthly patterns of wind, precipitation humidity and moisture transport in the Kenya-Tanzania border area. We concentrate on the months around the end of the respective rainy seasons in the area, when we may expect the largest effect of greening, as

5 argued before. For the 'long rains' (March-April-May, MAM) we analyse the May and June months in Figure 8. For the 'short rains' (October-November-December, OND) we analyse December-January in Figure 9.



**Figure 8: 'Long rains' transition period: maps on top row of monthly rain (mm/day, contours) and wind (quivers) on a background of land elevation (topography, m amsl, colours) and on the bottom row of relative humidity (%, contours) and**
10 **atmospheric moisture transport (quivers) on a background of precipitation (mm/day, colours). Left for May, right for June. The red 'K' indicates the location of Mt Kilimanjaro.**





Figure 8 shows that in the May-June period winds from the south east dominate the region. The winds are deflected and channelled by the topography to more southerly winds over Kenya, turning towards the west again over the Marsabit depression, and more winds from the east flow over Tanzania. In May this leads to coastal rainfall concentrated north of Mombasa. In addition orographic effects induce relatively high rainfall amounts around the Mt Kilimanjaro and Mt Kenya

5 massifs respectively and again west of the Rift Valley. In June the picture is similar though moisture transport inland is already strongly diminished and relative humidity is 10% lower.

**Figure 9: 'Short rains' transition period: maps on top row of monthly rain (mm/day, contours) and wind (quivers) on a**
10 **background of land elevation (topography, m amsl, colours) and on the bottom row of relative humidity (%, contours) and atmospheric moisture transport (quivers) on a background of precipitation (mm/day, colours). Left for December, right for January.**



In December-January (Figure 9) the wind comes from north-easterly directions. Although topography still concentrates rain on its windward sides, this effect seems reduced compared to the long rains transition in May-June. Rather dry air masses penetrate the area from the north after the short rainy season.

From this analysis it follows that during the transition after the long rains the Tanzanian corridors probably have the largest effects on rainfall, as winds are flowing from the south-east bringing moisture directly from the ocean, especially in May. Following the short rains, the Kenyan corridors probably may have the largest effects on rainfall, as winds are flowing from the north-east. However, as these pass more over land than in May-June these winds are already drier and the potential for greening-enhanced rainfall may be reduced compared to the May-June period.

## 5.3 Model sensitivity of the May-June precipitation to greening

We use a regional weather model (RAMS v6.1a3, Cotton et al. 2003, Pielke et al. 1992) coupled to a land-surface model (LEAF3, Walko et al. 2000) to assess the potential effects of a greener vegetation on rainfall. We focus on the end of the 'long rains', performing simulations for the period 15 May-30 June. To provide lateral conditions for our model we used ECMWF re-analysis data for June 1994, a more or less average year in terms of annual precipitation.

A series of simulations was performed with

1. Present day vegetation (base line)
2. Greener vegetation over the entire area of each of the four Hydrological Corridors

Greener vegetation has been effectuated by upgrading most pixels in these areas to the next greener predefined land cover class (in the LEAF3 land surface model integrated in RAMS), which amounts to an increase of vegetation Leaf Area Index (LAImax) as listed in Table 3.

**Table 3: Assumed upgrade of land cover classes through re-greening.**

| | | | | |
|---|---|---|---|---|
| Short grass | $(8, LAI_{mx} = 2)$ | -> | Tall grass | $(9, LAI_{mx} = 5)$ |
| Tall grass | $(9, LAI_{mx} = 5)$ | -> | Wooded grasslands | $(18, LAI_{mx} = 6)$ |
| Wooded grasslands | $(18, LAI_{mx} = 6)$ | -> | Mixed Woodlands | $(14, LAI_{mx} = 7)$ |

The vegetation model uses these maximum LAI values in a fixed phenological cycle to come to a time varying LAI throughout the seasons. For the period that was simulated, this led to modest increases in LAI and decreases in albedo as shown in Table 4. Also surface roughness marginally increased.



**Table 4: Realized changes in surface characteristics of the four sensitivity experiments.**

| Greened corridor | Simulation number | Sample area | Effective LAI | | Effective albedo | |
|---|---|---|---|---|---|---|
| | | | baseline | green | baseline | green |
| Ke-E | ng122 | 38.2-39.2E; 3.5-4.0S | 1.02 | 1.54 | 0.18 | 0.15 |
| Ke-W | ng112 | 35.2-36.2E; 1.2-1.7S | 1.08 | 1.55 | 0.19 | 0.16 |
| Tz-E | ng102 | 37.5-38.5E; 4.0-4.5S | 1.01 | 1.45 | 0.18 | 0.16 |
| Tz-W | ng92 | 35.0-36.0E; 2.5-3.0S | 0.70 | 1.20 | 0.22 | 0.21 |

Figures 10 to 13 are maps of the difference in precipitation between the baseline simulation and each of the greened corridors (Figures 10 and 11 for Tanzania, and Figures 12 and 13 for Kenya), together with maps of the associated primary

5  effects that may affect precipitation, namely the changes in surface heat flux and especially evaporation and their effect on temperature and humidity. In Figures 10 and 12 we present the baseline situation (left column) and the effects of greening in the West and East corridors (middle and right column, resp.), given as the difference from the baseline, on evaporation (IE, 1st row) and heat flux (H, 2nd row); percentage relative humidity (RH, 3rd row) and temperature in Celsius (T, 4th row). Blue colours indicate moister or cooler conditions, red colours dryer or warmer conditions (but note there are different

10  colour scales for each variable). Figures 11 and 13 show the changes in precipitation.

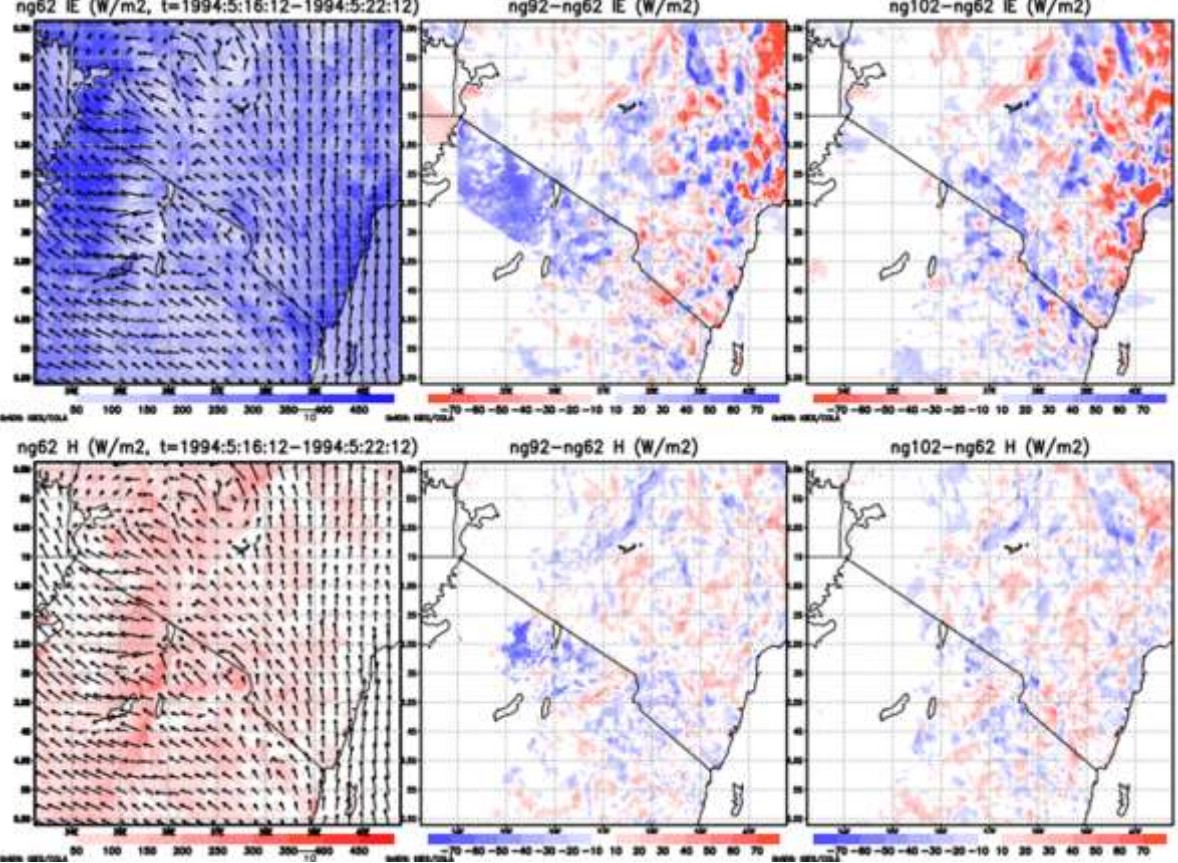





**Figure 10: Effects of greening the two Tanzanian corridors during the first week of the simulation (16-22 May1994). The middle and right column show the effects in the West and East corridors resp., given as the difference from the baseline in the first column, on evaporation (1st row) and heat flux (2nd row), both are in W/m2 (680 W/m2 evaporation roughly equals 1 mm/hr); percentage relative humidity (3rd row) and temperature in Celsius (4th row).**




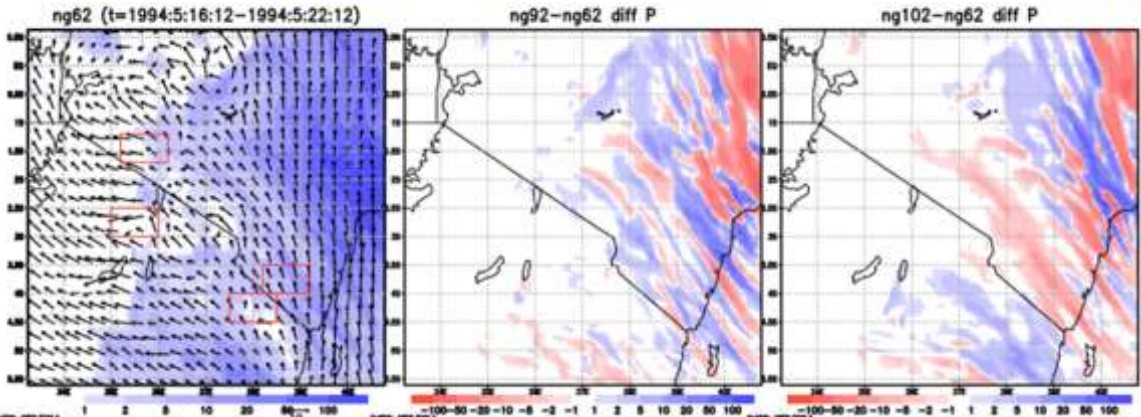

**Figure 11: Changes in precipitation in the Tanzanian West and East corridors for the same period as in figure 10. Note the non-linear colour scale.**

For the two Tanzanian corridors (Figure 10) greening leads to an increase of evaporation in the first week of the simulation,

especially in the western corridor (from 141 to 185 W/m2 or 31%) and an associated but smaller decrease in heat flux (due to the albedo increase more solar energy is absorbed by the vegetation). This leads to a slight increase of relative humidity in each area, and similar changes elsewhere in the domain (for which we don't have a clear explanation yet). Temperature changes are marginal. Precipitation in the eastern part of the domain responds rather sensitively to these surface changes (Figure 11): in the north-eastern area there is a wave like pattern of rainfall increases and decreases that confounds the

effects of the greened corridors. Nevertheless and considering also the average wind field for this week the graphs suggest that greening the western corridor in Tanzania has no effect on rainfall anywhere in the domain (middle graph). Greening the eastern corridor (on the right) seems to enhance rain inside this corridor, but possibly at the same time reduces rain downwind across the Kenyan border: in the two right hand side rectangular areas the rain increased from 2 to 4 mm in the Tanzanian box, but decreased from 8 to 3mm in the Kenyan box. The greening may have triggered rain initiation in Tz-E but

after having rained out less moisture remained for rain in Ke-E.

In the following weeks the pattern changes (no maps are provided here, see Hutjes et al., 2016). The extra evaporation in the greened areas decreases and turns into reduced evaporation in the greened areas in week 4 – 6, especially in the Tz-W corridor. This is probably because the enhanced evaporation in the weeks before depletes soil moisture faster than in the baseline situation. As a result relative humidity over the greened areas is slowly reduced to below that of the baseline

situation while the temperature increases. Unfortunately there is little inland rain in the subsequent weeks, though in week 4 greening the eastern corridor suggests to enhance rain a bit downwind in Kenya and in week 6 it decreases the rain a bit over the corridor itself. As the wet season dwindles and the dry season strengthens the inland atmosphere quickly dries out to the point that surface conditions cannot affect rainfall anymore. In this period, the atmosphere remains moist enough for rainfall to occur only over the coastal areas and thus only in this area the land surface can affect the rainfall.



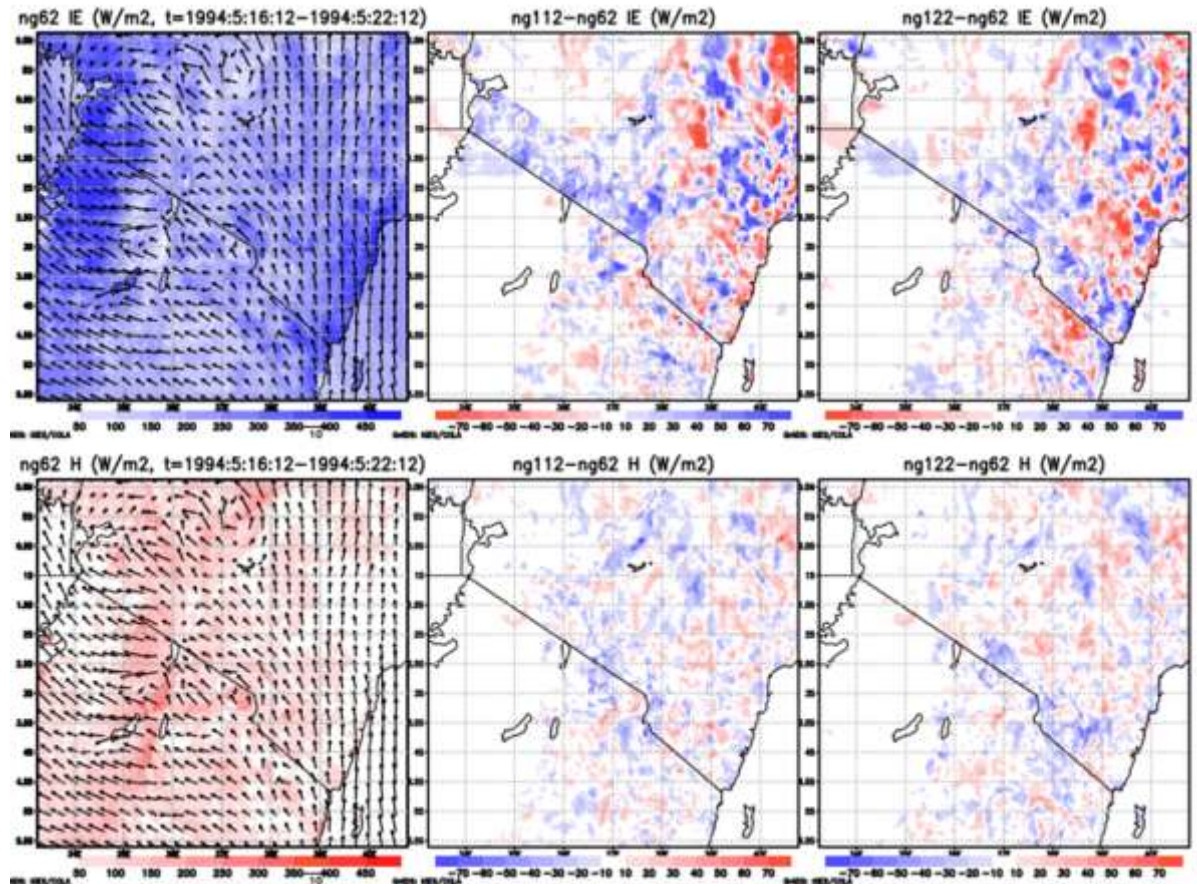





**Figure 12: Effects of greening the two Kenyan corridors during the first week of the simulation (16-22 May 1994). The middle and right column show the effects in the West and East corridors resp., given as the difference from the baseline in the first column, on evaporation (1st row) and heat flux (2nd row), both are in W/m2 (680 W/m2 evaporation roughly equals 1 mm/hr); percentage relative humidity (3rd row) and temperature in Celsius (4th row).**



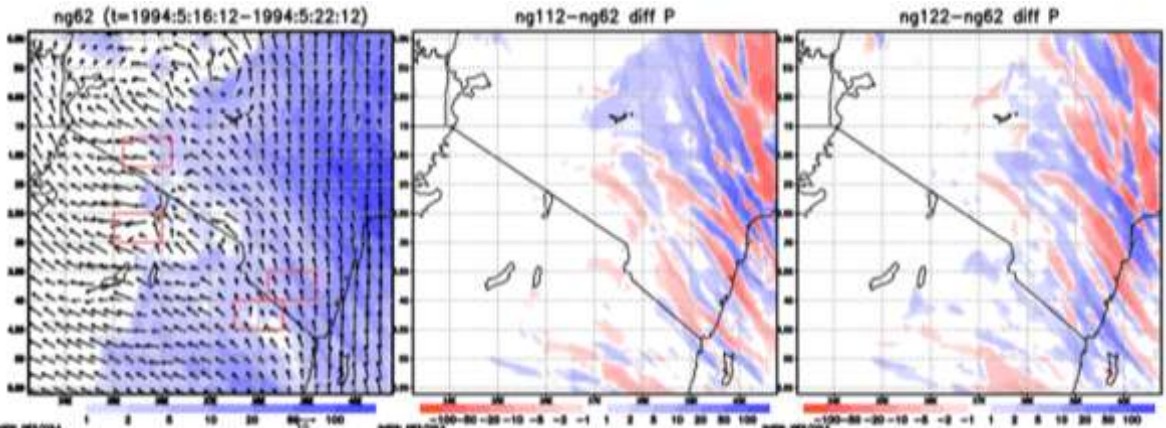

**Figure 13: Changes in precipitation in the Kenyan West and East corridors for the same period as in figure 12. Note the non-linear colour scale.**

For the two Kenyan corridors the effects are very similar, though the effects on rainfall are even more inconclusive (Figure 12 and Figure 13).

The sensitivity experiments do not show a very distinct rainfall feedback due to greening; however, in this particular simulation period, the experiments do demonstrate the land-atmosphere complexities involved. Greening not only leads to more evaporation through leaves, but also to absorption of more solar energy that –if not used for evaporation- will heat up

the air. More evaporation also quicker depletes available soil moisture stores, which -if these are not replenished- leads to reduced evaporation compared to the baseline as time moves on. In the atmosphere (more in particular, in the boundary layer, i.e. the first 1-2km or so) rain initiation depends on a subtle balance between enough moisture and enough buoyancy to bring that moisture to condensation levels. Surface conditions, evaporation and sensible heat flux may affect both factors and the overall effect depends which factor is affected the most in the direction of promoting rainfall.

Some model limitations may have reduced a very clear feedback of greening on rainfall. The most prominent limitation is the fact that the model has a static vegetation development that does not respond to more rain or more soil moisture by postponing senescence. In our model the seasonal cycle is fixed, so even when soil moisture would permit otherwise, leaf area reduces as the season progresses. A so-called interactive vegetation model might have strengthened feedbacks.

For the same reason the imposed greening, though extensive in area, enhances vegetation with a limited magnitude only

(LAI increased by 0.5 to still low values of 1.5 on average only, Table 4). With stronger greening the effects may also have been stronger. However, whether that level can be achieved for the full corridors remains to be seen.





## 6. The institutional context in Kenia and Tanzania

The Hydrological Corridor as promoted by Naga is to be realized by re-greening and restoration of the landscape. For the selection of a proper location it is of eminent importance that the changes are realized over a large area, are long lasting and are supported by the local population and future managers. This is not automatically the case as the Naga initiative originates

from outside the community and projects can hardly be realized without the agreement of higher administration: a basic conflict between top-down and bottom-up would arise. Therefore, the project should be developed in a way that fits with available institutions, has the support of governments as well as the local populations, builds on existing aspiration levels and makes use of available resources as much as possible.

For a prioritisation of project areas a comparison of these social and economic aspects is necessary. In this section we assess

the institutions at national level to check if those institutions would or would not support such an experiment. The term institutions is used in the broad definition of social rules governing (a part of) society (IDGEC 1999). The assumption is that the potential corridor with the most positive outcome (the most adaptive and most effectively governed location) would be the best option for the experiment. One could also reason that the location with the largest social and institutional problems mostly needs such an experiment; however, we reason that an experiment can fail, and a very vulnerable location can carry

the burden of such a failure even less.

### 6.1 Method: Adaptive capacity wheel assessment

We use the Adaptive Capacity Wheel (ACW, Gupta et al., 2010) that was developed to assess if adaptive capacity to climate change is enabled by social institutions. The problem in the Naga project is only partly related to climate change, but the set of criteria used in the ACW can nevertheless provide a relevant assessment. It basically measures both the structural

robustness of an institutional framework in terms of fair governance, resources and leadership, and at the same time the extent to which it enables people to adapt using criteria on variety, learning and room for autonomous change. The method checks institutions along 22 criteria, clustered in six dimensions. For each criterion a score is given between -2 and +2 and an explanation to argue the score. Then a simple calculation method is used to aggregate over the dimensions and to arrive at the overall score.

We first aimed to assess the four different regions identified before: Tanzania East / West and Kenia East / West. This requires detailed information on cultural differences between the peoples inhabiting those four areas. There are over 120 different ethnic groups among Tanzania's population and over 70 distinct ethnic groups in Kenya (some of which overlap) and this would lead to a very complex analysis. Moreover, the information where each ethnic group lives exactly is sketchy and sensitive. Therefore, it was decided to only look for differences between the institutions at the national level in the two

countries. Of course cultural differences within the countries are also very relevant (for example, differences between cattle herders and crop farmers) and this will have to be investigated further in a next phase of the project.





The data used are from literature. In one interview with a Dutch person who runs a medical service in Tanzania feedback was asked on a first draft of this document. For each country a table with the criteria is filled with data, consisting mostly literal citations from the available literature (for the original tables see Hutjes et al., 2016). Validity of the results needs to be checked locally, as we had to judge it from a distance with limited and possibly outdated information.

## 6.2 Results of the Adaptive Capacity Wheel assessment

The results of the analysis with the Adaptive Capacity Wheel are summarized in Table 5. For Kenya data were more difficult to find, and literature on sub-Saharan Africa was used to fill a part of the gaps. In Kenya the ACW dimension 'variety' is high, not only because of ethnic diversity but also because many different solutions are tried and because many stakeholders are involved in the climate response strategy. However, learning is seriously hampered by a lack of trust due to violence between ethnic groups. One would expect the room for autonomous change to be wide in a country where people are left to their own resources, but in Kenya people are severely limited in that respect due to their dependence on rain-fed agriculture. The freedom to fall back on illegal activities cannot be considered an advantage. The score for 'act according to plan' is quite positive based on a government report (Government of Kenya, 2013); however, the validity of such a self-assessment is questionable. If we go to the structural governance dimensions the situation looks problematic: leadership is scoring rather low and fair governance has the lowest possible score. The resources are mostly lacking even though the economy is growing and a new climate fund is considered.

For Tanzania the data are more comprehensive than for Kenya due to a recent, rather critical analysis of resilience to climate change (Hepworth 2010). When we look at the dimensions describing the adaptive, flexible side of governance, variety is again high because of cultural diversity, involvement of stakeholders in adaptation and experimentation with many different solutions. The redundancy criterion scores low because many people hardly have a margin or a buffer for securing their survival. Learning shows some promise due to investments in research but this is mainly based on foreign aid. In Tanzania there has been a major effort towards literacy, even though much remains to be done. Room for autonomous change scores low because the people of Tanzania have no access to data, there is a lack of implementation and there are general poverty issues. When we look at the dimensions that reflect structural governance, the leadership has positive scores due to the emphasis on the Tanzanian national identity at the expense of tribalism. Resources are problematic, especially due to a fast-growing, uneducated population, but authority shows some positive scores. Although population growth can be seen as good for human resources, proper education of the youth would be needed for enhancement of adaptive capacity. Financial resources coming from foreign donors seem necessary for planning of adaptation. Even though the economy of Tanzania is growing, the revenues are not channelled towards the common good due to a lack of fair governance. Fair governance is weak but on some criteria positive developments are reported such as strategic support for vulnerable groups. Tanzania is working to improve fair governance in a structural way, for example, by trying to change the position of women and by reducing corruption.



**Table 5: Results of the Adaptive Capacity Wheel assessment (for annotated results see Hutjes et al., 2016).**

| Dimension | Criterion | Score Kenya | Score Tanzania |
|---|---|---|---|
| **Variety** | Variety of problem frames & solutions | +1 | +2 |
| | Multi-actor, level and sector | +2 | +2 |
| | Room for diversity | +2 | +1 |
| | Redundancy | -2 | -2 |
| | **Total** | **+0.8** | **+0,8** |
| **Learning Capacity** | Trust | -2 | 0 |
| | Double loop learning | -1 | +1 |
| | Discuss doubts | 0 | +2 |
| | Single loop learning | 0 | -1 |
| | Institutional memory | +1 | -1 |
| | **Total** | **-0,4** | **+0,2** |
| **Room for autonomous change** | Continuous access to information | -1 | -1 |
| | Act according to plan | +1 | -2 |
| | Capacity to improvise | -1 | -2 |
| | **Total** | **-0.3** | **-1,6** |
| **Leadership** | Visionary leadership | -2 | -1 |
| | Entrepreneurial leadership | 0 | +2 |
| | Collaborative leadership | -2 | +2 |
| | **Total** | **-1** | **+1** |
| **Resources** | Authority | -2 | +1 |
| | Human resources | -2 | -2 |
| | Financial resources | 0 | -1 |
| | **Total** | **-1.3** | **-0,6** |
| **Fair Governance** | Legitimacy | -2 | 0 |
| | Equity | -2 | +1 |
| | Responsiveness | -2 | -2 |
| | Accountability | -2 | -1 |
| | **Total** | **-2** | **-0.5** |
| **Overall** | | **-0.7** | **+0,1** |

## 7. Conclusions

### 7.1 Reflection on the method

The aim of this paper was to support an objective choice between potential intervention projects in which the Naga
5  Foundation could establish Hydrological Corridors for re-greening and restoration of the landscape. We collected
geographically explicit information on soil, vegetation, hydrology and climate, as well as institutional settings, which may all



affect the likelihood of success of such projects. Four corridors were pre-selected by the Naga Foundation, all close to the Kilimanjaro region: Kenya-East, Kenya-West, Tanzania-East and Tanzania-West.

Each of the analyses in previous sections used different criteria to assess which of the four corridors would be most suitable for starting re-greening projects. The information gathered on hydrological and soil restoration potential was available at a very fine scale. The GIS data made available through maps will allow a more detailed search for suitable project locations. The geographical level of detail of the climatological and institutional analysis is lower but still provides useful information on how to deal with projects in these four corridors.

In the following subsections conclusions are presented for each domain, and finally in section 7.6 the four assessments are combined in an overall outcome regarding the choice for a Hydrological Corridor.

## 7.2 Concluding remarks on soil restoration

Regarding the aggregate scope and effective potential for soil organic matter or carbon (SOC) Ke-E has a high overall scope, but a relatively low potential for effective restoration using SLM measures. Tz-W and Tz-E also have a quite high scope for SOC restoration, and in terms of achievable restoration potential using the SLM measures considered they feature the highest potential, especially Tz-E. Ke-W scores lower on both accounts. For planning the interventions in more detail, a more in-depth assessment may be needed which SLM measures are possible in what locations and it could be useful to revisit the assessment with higher resolution data. Similarly, the assessment could be repeated for specific technologies rather than generic categories of SLM measures. Furthermore, the method should be adapted somewhat for national parks as not all SLM measures will be applicable there.

## 7.3 Concluding remarks on hydrology and erosion

Regarding the hydrological characteristics, the potential for soil moisture and groundwater recharge seems highest in Tz-E where more sandy soils promote infiltration of the available rain.

## 7.4 Concluding remarks on rainfall feedbacks of greening the land

From a positive rainfall feedback perspective the highest chances for realizing an enhanced small/local hydrological cycle seem to exist in greening the Tz-E corridor. Following the long rains, wind directions and advected moisture from the oceans are strong enough for additional evaporation from the land surface to positively influence rainfall at least windward of Kilimanjaro. In this same season greening the Ke-E corridor will also create extra evaporation, but downwind of this area the topography is going down and thus the chances of this extra evaporation leading to extra rain are smaller. The westerly corridors Ke-W and Tz-W seem too far inland for greening to enhance rainfall. The analysis suggests the atmosphere is too dry here for greening to have an effect.



Inferring from the climatological analysis only, it seems that atmospheric conditions in the short rains season are less favourable for rainfall enhancement, although it cannot be excluded. If we would consider the short rains, the Ke-E corridor may be the more favourable one, since with winds from the north east it is then upwind from the rise in topography.

### 7.5 Concluding remarks on adaptive capacity

Tanzania and Kenya have many similarities in their social and economic context. Many cultural groups such as the Masaai can be found on both sides of the Kenyan-Tanzanian border. Regarding the adaptive capacity provided by the social institutions at the national scale it was concluded that both countries have positive scores on the dimension of variety, neutral scores for learning, and negative scores for availability of resources. On the dimension room for autonomous change Kenya scores higher than Tanzania but this may be due to better (more critical) data on the situation in Tanzania. Tanzania scores higher on leadership and fair governance. Overall Tanzania seems the better option for a large-scale, long-term experiment with reforestation as the government at the national level seems more reliable.

### 7.6 Combination of the four assessments in one framework

In 7.6 Combination of the four assessments in one framework

6 we combine the findings of the four assessments by ranking (1 lowest to 4 highest) the four potential corridors. These findings favour the Tanzanian corridors and especially the Tz-E one, to start with re-greening projects. In that region many applicable land management options combine with a high potential for restoring soil organic matter, the highest rainfall recycling potential exists in the more favourable long rains season, while the Tanzanian government at the national level seems more reliable. The GIS data facilitate further zooming in on this particular corridor in search of specific project locations, especially with respect to hydrological and soil restoration characteristics for which highly detailed information is available. The institutional setting could be further analysed with additional data on local communities and local cultural views on land use and re-greening measures.

**Table 6: Cross-theme ranking (1 lowest to 4 highest) of the four potential corridors.**

|  | Water infiltration potential | Realistic SOC restoration potential 2020 | Absolute soil restoration potential | MAM rainfall feedbacks | OND rainfall feedbacks | Institutional potential | **Total score** |
|---|---|---|---|---|---|---|---|
| Kenya-East Corridor | 2.5 | 2 | 1 | 3 | 4 | 1.5 | **14** |
| Kenya-West Corridor | 1 | 1 | 4 | 1.5 | 1.5 | 1.5 | **10.5** |
| Tanzania-East Corridor | 4 | 4 | 3 | 4 | 3 | 3.5 | **21.5** |
| Tanzania-West Corridor | 2.5 | 3 | 2 | 1.5 | 1.5 | 3.5 | **14** |
| Total | 10 | 10 | 10 | 10 | 10 | 10 | 60 |



**Acknowledgements**

For initiating and funding the described research, we acknowledge the support of the Naga Foundation.

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





## Appendix A

Classification of Sustainable Land management (SLM) categories

| SLM Category number | SLM Category name (and acronym) | Description |
|---|---|---|
| 1 | Afforestation (AFFOR) | Planting of trees or bushes for regeneration of natural areas or reclaiming degraded land. Afforestation is sometimes combined with other measures such as terracing, and is often initiated especially to reduce off-site effects of land degradation (including wind erosion). |
| 2 | Agroforestry (AGROF) | Agroforestry entails the integration of trees or tree crops and agriculture for synergies in productivity and ecosystem service provisioning. From a SLM perspective interventions are concerned with better soil cover and increased production. |
| 3 | Agronomic measures (AGRON) | Agronomic measures cover a broad range of options that have in common that they need to be annually repeated and/or have a short-term effect. |
| 4 | Bunds (BUND) | Bunds are defined as structural cross-slope measures that can be constructed without a massive amount of earth movement (which is characteristic for terracing). Over time, slopes treated with bunds can gradually form into terraces. Their main objective is to break the slope to reduce the velocity of run-off and increase infiltration. |
| 5 | Grazing land management (GRZMGT) | This category includes management strategies for grazing land and rangelands in drylands. Their main purpose is to facilitate the build-up of vegetation and soil organic matter. The environments covered by this category include grasslands and shrub lands. |
| 6 | Gully rehabilitation (GULREH) | The measures in this category serve to mitigate gully development or rehabilitate degraded lands. Frequently, the aim is to reduce flow velocity and/or protect gully banks and heads. |
| 7 | Home garden improvement (HOMGAR) | Home garden improvement measures are often combinations of multiple technologies requiring high or frequent resource inputs, especially of labour. They are thus often relatively expensive and usually confined to small-scale intervention areas. |
| 8 | Irrigation (IRRI) | Irrigation involves developing permanent access to water for increased productivity and/or increased water use efficiency of existing irrigation systems. Irrigation technology is often expensive but leads to large productivity increases, out of season cropping opportunities and increased resilience to droughts. |
| 9 | Terracing (TERRA) | Terracing reduces slope length and slope gradient by substantial earth movement, sometimes accompanied by stabilisation measures (stone walls, vegetation). |





| | | |
|---|---|---|
| | | Construction of terraces is labour (or machine) intensive. Apart from controlling soil erosion, terraces may also achieve water conservation, and offer opportunities for intensification or increased productivity of agricultural systems. Terraces can enable agriculture in areas otherwise unsuitable, or allow introduction of irrigation. |
| 10 | Vegetative barriers (VEGB) | Vegetative barriers refer to cross-slope planting of vegetation as a permanent barrier. This intervention breaks the slope, thereby reducing velocity of run-off and increasing infiltration. Vegetative barriers are often combined with bunds, and the distinction is arbitrary to a certain extent. |
| 11 | Vegetative cover (VEGCOV) | Measures in this category aim at the (re-)establishment of vegetative ground cover to reduce the impact of rainfall and/or kick-start regeneration. The cover is sown and consists of annual crops, grasses and/or shrubs. Cover crops can be implemented to protect against soil erosion, but commonly also provide other services such as fodder and nitrogen fixation. |
| 12 | Water harvesting (WATHAR) | Water harvesting increases availability of water through collecting, conveying and storing water, and is often applied when (in situ) precipitation is not sufficient to meet water requirements. A wide variety of water harvesting systems exist, some of which are applied within fields, while others capture water ex-situ. Water harvesting differs from irrigation in the sense that there is still a dependency on precipitation. |
| Re-forestation measures | Afforestation of savannahs (AFSAV) | For the location search for hydrological corridors in Kenya and Tanzania, the option afforestation of savannahs is also relevant. This measure covers reforestation of areas in the savannah biome that are not under forest cover and agriculture. |