# Peer review of "Hydrological corridors for landscape and climate restoration: Prioritization of re-greening areas in Kenya and Tanzania"

_SOIL, 2016_

## Referee Comment (RC1) · Anonymous Referee #1 · 25 May 2016

This manuscript describes the selection process used to identify land areas in Kenya and Tanzania that might support "re-greening" efforts in the development of "hydrological corridors" through the use of "sustainable land management" (SLM) practices, such as afforestation, gully rehabilitation, irrigation, terracing, and so forth. The greater goal of the effort is to assist a non-profit organization from The Netherlands in combating dryland desertification in East Africa. The authors draw largely on existing environmental data to create a GIS-based assessment of which land area(s) might be most suitable for the proposed project. The authors also examine national-level social and economic data to consider possible institutional support for the project. This is an interesting study (although the English language writing needs attention). I have four

overall comments about this manuscript:

1. The authors assume that the land areas under consideration are "degraded," but do not demonstrate this empirically nor define what they mean by "degraded." The authors also assume that the land areas under study were once "green" (but do not explain what this means; perhaps "not degraded"?) and that these land areas need to be (and should be) "re-greened." Moreover, the authors assume that the people living in these land areas desire to have their lands "re-greened" and to have it done by people from outside their country and culture. These are some hefty assumptions. I think it's necessary for the authors to identify (make explicit) these assumptions and defend them.

2. The authors write that they "used a number of objective criteria for prioritizing potential areas for re-greening interventions," but the culmination of the analysis is a set of subjective rankings used to identify "desired characteristics" by giving high scores to some criteria and low scores to "less favourable ones." The result is a subjective assessment of what the authors feel is "best" for the success of SLM practices. This is ok to do, but the authors need to be up front about this and identify this process as a limitation of the study.

3. The "adaptive capacity wheel assessment" is especially subjective and not culturally sensitive. Moreover, the methods and interpretation of the results (including Table 5) are unclear. While I think it's very important to assess institutional capacity, the application of this particular kind of assessment seems rather biased and even ethnocentric. I think its inclusion in this manuscript weakens the paper.

4. Where do the data come from that were used to create Figures 2-6? How were these images created? This is especially important for Figures 3-6, which are assessments (not measurements). Also, the labels for each of the land areas only appear in Figure 6, and so the authors' observations throughout the manuscript are nearly impossible to follow.

---

## Referee Comment (RC2) · Anonymous Referee #2 · 22 Jun 2016

General comments: Contribution is focused on problems of restoration in Kenya and Tanzania. More accurately it is concentrated on evaluation and prioritization of different areas, which are candidate for restoration. System for evaluation includes soil, water, climate and social institutions also. In this contribution I mostly appreciate, that authors have not only focused on climate in a global scale but on a local scale as well. Moreover, text contains clear description causes of desertification in the sense of direct and indirect way. To my mind authors have prepared useful system for evaluation and for choosing areas which represent localities with more or less same weight. Creating this system looks like time-consuming and authors should seek for a clarification of this system Specific comments: On pages number 5, line 15-20, I can see the information

about enhancing soil health using amendments or altering physical and biological properties. You discuss the use of manure or compost for enhancing chemical fertile soil and for improving soil cover. I am missing the information how you can alter biological properties of soil.

———————————————————

---

## Author Comment (AC1) · 23 Jun 2016

Reply on comments of Referee 1

1. The authors assume that the land areas under consideration are "degraded," but do not demonstrate this empirically nor define what they mean by "degraded." The authors also assume that the land areas under study were once "green" (but do not explain what this means; perhaps "not degraded"?) and that these land areas need to be (and should be) "re-greened." Moreover, the authors assume that the people living in these land areas desire to have their lands "re-greened" and to have it done by people from outside their country and culture. These are some hefty assumptions. I think it's necessary for the authors to identify (make explicit) these assumptions and

defend them.

1a. The authors assume that the land areas under consideration are "degraded," but do not demonstrate this empirically nor define what they mean by "degraded."

We define 'degraded' in general terms first; the introduction mentions that: "A complex interplay of resource exploitation, population increase and environmental change is driving land degradation processes with both local and off-site impacts that may culminate in desertification, defined as the loss of productive capacity of drylands (UNCCD, 1994)". We can make this more clear by adding the word 'define' to this sentence. And in 3.2 the proxy is explained: "Degradation can be assessed by the soil depth loss and the carbon content of the soil relative to undisturbed conditions." The degree of degradation is shown by the annual soil depth loss in figure 2 and the restauration potential SOC levels in figure 3. Apparently not the whole area is degraded at to the same degree, and consequently greening interventions will take place at -but are not confined to- those degraded locations. Empirical work is not included in the study; for this the consent of the local people would be necessary and that is one of the next steps. This article only describes how to select locations where it can be useful at all.

1b. The authors also assume that the land areas under study were once "green" (but do not explain what this means; perhaps "not degraded"?).

Of course there is a logical link between 'green' and 'not (yet) degraded', but degradation is only one of the project concerns. The wider goal of the project is moderation of climate extremes (hot, dry, downpours) through greening of the landscape in order to create a Hydrological Corridor. Greening then means to bring the vegetation to a higher level of evapotranspiration. Therefore the term 'vegetation degradation' has been introduced in line 6.

1.c ....and that these land areas need to be (and should be) "re-greened." Moreover, the authors assume that the people living in these land areas desire to have their lands "re-greened."
Re-greening indeed is the intended activity that will have to take place in sufficiently large and well chosen areas. This gives the NGO an inspiring mission which not automatically is shared by all. In order to realize sustainable development over such a large scale support at all levels must be acquired; a reason to include the chapter Institutional context in the study. Re-greening only will be successful if an increase in biodiversity and land productivity can be realized with an appropriate level of stakeholder and government support. We can explain this more explicitly in the introduction.

1.e ".......and to have it done by people from outside their country and culture. These are some hefty assumptions."

It is not the intention to implement the programme in East Africa without communities from inside the country. It is the intention to come with concrete proposals for improvement of local climate in areas where regreening can have the most significant impact. The method is developed not only for East Africa but for long term actions worldwide. It is a way to put modeling and data into practice to achieve climate adaptation. Requirements for good governance are part of the institutional analysis and these apply to the local stakeholders as well as to the international partners. We can explain this more explicitly in the introduction.

1.f "I think it's necessary for the authors to identify (make explicit) these assumptions and defend them."

Where required for a better understanding the text of the study, adjustments will be made. Information on the more general nature of the project will be added in the introduction.

2. The authors write that they "used a number of objective criteria for prioritizing potential areas for re-greening interventions," but the culmination of the analysis is a set of subjective rankings used to identify "desired characteristics" by giving high scores to some criteria and low scores to "less favourable ones." The result is a subjective assessment of what the authors feel is "best" for the success of SLM practices. This is

ok to do, but the authors need to be up front about this and identify this process as a limitation of the study.

Referee 1 is completely right. We did select objective criteria – but this selection is already subjective - and to fill the gaps towards a final recommendation we used subjective mechanisms as well, especially in the final analysis; however we tried to be as explicit as possible about all the steps we made. We will adjust this in the text.

3. The "adaptive capacity wheel assessment" is especially subjective and not culturally sensitive. Moreover, the methods and interpretation of the results (including Table 5) are unclear. While I think it's very important to assess institutional capacity, the application of this particular kind of assessment seems rather biased and even ethnocentric. I think its inclusion in this manuscript weakens the paper.

This opinion of Referee 1 will partly be caused by the absence of the data on which this analysis is based; it may look like the scores were added to the criteria out of the blue. The table with supporting data for this analysis was left out because it added too many words to the article, but it can be added as supplementary material.

This analysis is undeniably ethnocentric and culturally biased; however, this applies to all scientific research that is published, natural science as well as social science. Then every article should start with a caveat: "We have to warn the reader that, we, the researchers, are white / coloured Europeans / Americans, who believe that models and data are relevant to understand the dynamics of planet Earth." For the analysis we have used material that was available on the internet; it included government reports from the region. Interviews on the ground would have provided a more diverse view but collecting field data was not included in the investigation (for any of the domains).

As was explained under point 1, we see the inclusion of this part in the study exactly as the opposite: it is a way to introduce cultural sensitivity from the start by including the people and their opinions in the analysis; even if it is with insufficient data at this stage.

4. Where do the data come from that were used to create Figures 2-6? How were these images created? This is especially important for Figures 3-6, which are assessments (not measurements). Also, the labels for each of the land areas only appear in Figure 6, and so the authors' observations throughout the manuscript are nearly impossible to follow.

No response to this as our co-author is out of the country; it will follow later.

Finally: "This is an interesting study (although the English language writing needs attention)."

Isn't it a bit ethnocentric to require of the whole world that they write English like a native English speaker? Please help and tell us what our mistakes are, because now we have to look for a needle in a haystack. Or at least indicate how many needles you found.

---

## Short Comment (SC1) · 28 Jun 2016

4. Where do the data come from that were used to create Figures 2-6? How were these images created? This is especially important for Figures 3-6, which are assessments (not measurements). Also, the labels for each of the land areas only appear in Figure 6, and so the authors' observations throughout the manuscript are nearly impossible to follow.

The reviewer is right to request more details on the data and procedures to create Figures 2-6. First of all, the fact that labels of the potential corridors only appeared in Figure 6 was indeed confusing. These have now been added to Figure 1 to better set the scene for the assessment from the start of the paper. Concerning the data and

procedures used to construct Figures 2-6, the first paragraph of Section 3.1 Method for soil restoration has been extended as follows: "For prioritizing regions for soil restoration, we take the existing level of degradation as a relevant first indicator. Degradation can be assessed by the soil depth loss and the carbon content of the soil relative to undisturbed conditions. Current soil organic carbon content, as well as ceiling values under agricultural land use and natural conditions were derived from the S-world global soil mapping approach (Stoorvogel, 2014). Ongoing degradation was determined by combining an assessment of vegetation cover degradation (Schut et al., 2015) and its effect on soil properties as modelled using S-world. For assessment of regreening potential, a number of land management interventions was considered. Different sustainable land management (SLM) interventions have different applicability limitations and their effectiveness also depends on environmental conditions. Spatial assessment of applicability limitations of SLM measures took into account land use, slope, rainfall, and soil depth and texture (cf. Fleskens et al., 2016). The number of options for recovery and their simulated effectiveness represent relevant indicators to include in our study."

Further in the same section, the sentences 'In order to calculate the effects, both restoration and prevention were expressed regarding their effect on SOC, which can be considered a proxy indicator for soil productivity. The extent to which they contribute depends on time.' were extended to read as follows: "'In order to calculate the effects, both restoration and prevention were expressed regarding their effect on SOC, which can be considered a proxy indicator for soil productivity. The initial values and maximum potential for both effects were informed by the S-World global soil modelling approach (Stoorvogel, 2014). The SLM measure-specific extent to which restoration and prevention potentials are reached depends on time."

In Section 3.2, the first sentence 'Figure 2 shows the annual soil depth loss in the research area, whereas Figure 3 shows the SOC restoration potential' was revised to read: "Figure 2 shows the annual soil depth loss in the research area, whereas Figure

3 shows the SOC restoration potential as determined with S-World simulations."

Figures and captions were revised accordingly to indicate areas and data sources.

References:

Fleskens, L., Kirkby, M.J., Irvine, B.J. (2016) The PESERA-DESMICE modelling framework for spatial assessment of the physical impact and economic viability of land degradation mitigation technologies. Frontiers in Environmental Science 4:31. doi: 10.3389/fenvs.2016.00031.

Schut, A.G.T., Ivits, E., Conijn, J.G., ten Brink, B., Fensholt, R. (2015) Trends in Global Vegetation Activity and Climatic Drivers Indicate a Decoupled Response to Climate Change. PLoS ONE 10(10): e0138013. doi: 10.1371/journal.pone.0138013

Stoorvogel, J.J. (2014) S-world: A global map of soil properties for modeling. In: Arrouays, D., McKenzie, N., Hempel, J., Richter de Forges, A., and A. McBratney (Eds). Global Soil Map: Basis of the Global Spatial Soil Information System, Taylor & Francis Group. pp. 227-231.

**Fig. 1.** Figure 1: Potential Hydrological Corridors in Kenya (Ke-W, Ke-E) and Tanzania (Tz-W, Tz-E) (imagery from Google Earth).

**Fig. 2.** Figure 2: Annual soil depth loss (mm/year) based on S-World simulations.

**Fig. 3.** Figure 3: SOC restoration potential (ton C/ha) based on S-World simulations.

[Figure]

**Fig. 4.** Figure 4: Number of applicable SLM categories.

**Fig. 5.** Figure 5: Total effective SOC restoration potential by 2020 (ton C/ha) realizable by applicable SLM measures.

---

## Short Comment (SC2) · 28 Jun 2016

General comments: Contribution is focused on problems of restoration in Kenya and Tanzania. More accurately it is concentrated on evaluation and prioritization of different areas, which are candidate for restoration. System for evaluation includes soil, water, climate and social institutions also. In this contribution I mostly appreciate, that authors have not only focused on climate in a global scale but on a local scale as well. Moreover, text contains clear description causes of desertification in the sense of direct and indirect way. To my mind authors have prepared useful system for evaluation and for choosing areas which represent localities with more or less same weight. Creating this system looks like time-consuming and authors should seek for a clarification of this

system

Reply: Thank you for your comment. Also in response to Reviewer 1, more details were added on how degraded areas were defined and how the partial assessments (water, soil, institutional capacity, climate) were made. We believe that with the added details further clarification is given. The reviewer's suggestion to comment on the usefulness of the system is a good one. To this effect, we will include the following statement in Section 7.1 Reflection on the method: "Although the integrated assessment method for selecting Hydrological Corridors requires a number of domain-specific assessments, these can be done simultaneously and can be based on available data, so that a first assessment of opportunities for regreening can be made remotely before start of work on the ground. This could make the method apealing for similar assessments in data-scarce environments."

Specific comments: On pages number 5, line 15-20, I can see the information about enhancing soil health using amendments or altering physical and biological properties. You discuss the use of manure or compost for enhancing chemical fertile soil and for improving soil cover. I am missing the information how you can alter biological properties of soil.

Reply: the reviewer is right that little information is presented on how to alter the biological properties of soils. As stated, the SLM measures considered (listed in Appendix A) would mostly affect these indirectly, e.g. by less intensive use of land or replenishing soil organic matter. We feel that it is beyond the scope of the paper to provide further details on this here as it would affect the balance of attention given to the different domains of the assessment system. Moreover, an assessment of how SLM measures affect biological soil properties would be difficult, as such information is usually not available in spatial (map) format.